# OpenIDS2: A low-cost, 3D-printed, open-source platform for reproducible construction of DNA microarray synthesizers

Junhyeong Kim☉, Haeun Kim☉, Duhee Bang ⑩*

Department of Chemistry, Yonsei University, Yonsei-ro, Seodaemun-gu, Seoul, Korea

☉ These authors contributed equally to this work and share first authorship.
* duheebang@yonsei.ac.kr

## Abstract

Oligonucleotide synthesis is a fundamental technology in various fields of life science, including synthetic biology, molecular diagnostics, and biotechnology. In this study, we present OpenIDS2, an open-source, second-generation inkjet-based DNA synthesizer designed to enable flexible and low-cost synthesis in laboratory settings. Built upon the original OpenIDS platform, OpenIDS2 reduces the overall device volume to approximately one-third, integrates all control electronics through a custom-designed printed circuit board (PCB), and improves operational stability and fabrication accessibility via a peristaltic pump–based bulk solution delivery system. Notably, most mechanical components are designed for 3D printing, allowing users to fabricate and assemble the system at low cost using widely available tools. This design significantly enhances reproducibility and global accessibility as an open-source hardware platform. The system supports phosphoramidite chemistry and successfully synthesized 15-mer poly(dT) sequences on CPG-based substrates, which were subsequently analyzed by urea-PAGE and HPLC. OpenIDS2 not only demonstrates the practicality of a fully open, benchtop oligonucleotide synthesizer, but it also serves as a reproducible and extensible foundational platform that lowers the barrier to entry for laboratory automation through its nature as an open-source contribution.

## Introduction

Oligonucleotides are essential materials used in a wide range of life science fields, including synthetic biology, molecular diagnostics, gene editing, and DNA-based data storage [1–4]. In particular, high-density DNA microarray technologies that enable precise and parallel synthesis of diverse sequences have emerged as foundational tools for these applications. Consequently, the demand for customizable, cost-effective, and high-density DNA synthesis platforms continues to grow.

**Data availability statement:** The authors confirm that all data underlying the findings are fully available without restriction. All files needed to construct the device are held at https://github.com/regiregire/OpenIDS2 All data for the paper presented within it.

**Funding:** This research was supported by the Pioneer Research Center Program through the National Research Foundation of Korea funded by the Ministry of Science, ICT & Future Planning (RS-2022-NR067569), and by the National Research Foundation of Korea (NRF) grant funded by the Korea government (MSIT) (RS-2024-00338316).

**Competing interests:** The authors have declared that no competing interests exist.

Various approaches have been developed for fabricating DNA microarrays, including photolithography [5–7], electrochemical synthesis [8], and inkjet-based methods [9–11]. Among these, inkjet-based synthesis offers significant advantages due to its non-contact, highly precise reagent deposition and high flexibility in both sequence design and positional control. Furthermore, because phosphoramidite chemistry can be directly applied, inkjet-based methods maintain compatibility with conventional oligo synthesis protocols. The technology also benefits from rapid innovation in the commercial inkjet printing industry, leading to improvements in printhead performance, reliability, and availability [12,13].

While commercial oligonucleotide synthesis services offer rapid synthesis of high-quality products, their business model presents significant limitations for certain research and development contexts. Ordering custom microarrays is often costly, and despite the fast synthesis time, the overall lead time is frequently extended by shipping and logistical delays. This makes the process impractical for studies that require rapid and iterative prototyping of new designs. This limitation also poses a significant barrier to research that aims to employ non-standard reagents or unconventional substrates. Finally, the need to transmit proprietary or sensitive sequence data to a third-party vendor raises valid concerns about data security and intellectual property protection.

The ability to fabricate DNA microarrays in-house presents several solutions to these challenges. It allows (1) low-cost and rapid iteration of custom arrays for specific research needs, (2) full control over the entire synthesis process for optimization and novel research, and (3) enhanced data privacy and security.

In recent years, the accessibility of open-source microcontrollers (e.g., Arduino, Raspberry Pi) and low-cost 3D printers has led to a growing number of open-source hardware projects for laboratory automation. Examples include Arduino-based smart home automation systems [14,15], programmable syringe pumps [16], automated chemical synthesis platforms [17], and open-source 3D-printed liquid handling workstations [18]. These developments have significantly lowered the barrier to building custom laboratory equipment and provided a foundation for accessible inkjet-based oligonucleotide synthesizers.

Building upon this foundation, we previously developed OpenIDS, the first open-source inkjet DNA synthesizer. It integrates an industrial printhead, open-source control system, and Python-based software to achieve phosphoramidite-based DNA synthesis [19]. The entire hardware and software stack was openly shared via GitHub, demonstrating the feasibility of constructing a lab-scale DNA microarray synthesizer.

OpenIDS had bulky syringe pumps for driving fluidics, complex wiring, and difficult assembly, which hindered its broader adoption in the research community. To improve upon the original platform, we developed OpenIDS2, a next-generation synthesizer designed to enhance reproducibility, user-friendliness, system compactness, and accessibility as an open-source platform. OpenIDS2 integrates a lightweight and fully 3D-printed mechanical structure, a custom PCB-based control system, low-cost fluid handling driven by peristaltic pumps, and a localized sealing mechanism for

maintaining a controlled synthesis environment. The system was designed to be easily built—even by users with limited fabrication experience—using readily available tools and components.

In this study, we describe the design and implementation of OpenIDS2. We also demonstrate its feasibility as a reproducible, scalable, and low-cost open-source DNA microarray synthesis platform for laboratory use.

## Materials and methods

### Synthesizer construction

The OpenIDS2 system was constructed using Polymethyl methacrylate (PMMA) panels and 3D-printed components. The top lid of the synthesis chamber was CNC-machined by a commercial PMMA supplier. The custom design includes threaded holes for component mounting and grooves for O-ring sealing. The remaining panels were cut using a laser cutting service and bonded with PMMA adhesive to form a rigid and transparent enclosure. If access to CNC machining or laser cutting services is limited, the chamber can alternatively be fabricated using 3D printing, although minor modifications may be required to ensure optical visibility and reliable sealing.

The synthesis chamber is composed of five bonded PMMA panels that form an airtight box. The top lid is fixed to the main frame and sealed with an O-ring. Manual support jacks are used to raise or lower the chamber vertically. When the chamber is lifted, the O-ring is compressed against the frame, creating a sealed environment.

Mounted on the top lid are a 5-head Xaar Irix printhead (Xaar, UK) array, five amidite ink vials, a camera, and a bulk solution nozzle holder. These components are secured using appropriately designed support structures. Inside the chamber, only a substrate holder and a linear stage are present. All electronic and mechanical components, including microcontrollers and pump systems, are located outside the synthesis environment to minimize contamination and reduce the internal volume.

To prevent evaporated components from the bulk solution from contaminating the amidites in the printheads, the inert gas inlet was positioned to blow gas from the printheads toward the bulk solution dispense position. Additionally, a waste pump was used to continuously aspirate the air from around the substrate holder. This setup established a flow of inert gas from the printhead toward the bulk solution dispense area.

### Fabrication of 3D-printed structural components

All structural components were designed using Fusion 360 software (Autodesk, USA). The main frame and mechanical support structures were printed using polylactic acid (PLA) filament on a Bambu Lab X1 Carbon FDM 3D printer (Bambu Lab, China). Components requiring chemical resistance, such as the substrate holder, bulk solution nozzle holder, and peristaltic pump housing, were printed using an LCD 3D printer (Phrozen Sonic Mighty Revo, Phrozen, Taiwan) with Liqcreate Strong-X or Deep Blue resin (Liqcreate, Netherlands).

The printed resin components were cleaned in isopropyl alcohol (IPA) for 5 minutes using an Elmasonic P-series ultrasonic cleaner (Elma, Germany). After cleaning, they were post-cured at 60°C for 2 hours using a Form Cure UV chamber (Formlabs, USA).

### Integrated control panel design and fabrication

The integrated control panel was designed using the EasyEDA platform (easyeda.com). A two-layer PCB was fabricated by a commercial PCB manufacturer (JLCPCB, China) using an FR4 substrate with a thickness of 1.6 mm, HASL surface finish, and 1 oz copper weight. Gerber files, a bill of materials (BOM), and pick-and-place data were generated from the design and submitted for fabrication. All components were assembled by the manufacturer using their PCB assembly (PCBA) service, resulting in a ready-to-use integrated panel. The board integrates key functionalities including power distribution, motor driver control, sensor input, and serial communication with Arduino-based microcontrollers. Detailed

circuit schematics, PCB layout files, and the full bill of materials are available in the project's GitHub repository (see Data Availability section).

## Reagents preparation

DMT-dT phosphoramidite was purchased from Sigma-Aldrich and dissolved to a final concentration of 0.25M. The activator, 4,5-dicyanoimidazole (DCI) was also obtained from Sigma-Aldrich and prepared at a concentration of 0.7M. Propylene carbonate (PC) used as a solvent was dried over molecular sieves prior to use to reduce residual moisture content. All reagents were handled within a chamber where humidity was maintained below 10%. To minimize exposure to moisture, the reagents were immediately loaded to the synthesizer after preparation. Additional reagents, including dichloroacetic acid (DCA) in dichloromethane (DCM, 3:97 v/v), acetonitrile (ACN) and oxidizer (0.2M iodine in water/THF/pyridine) were also purchased from DUKSAN (Korea). All chemicals were stored and handled according to standard anhydrous conditions to preserve their reactivity throughout the synthesis process.

## Oligonucleotide synthesis

The synthesis substrate, consisting of a CPG solid support preloaded with a 5′-linked dT, was mounted onto a custom 3D-printed substrate holder. The reaction chamber was sealed by lifting it upward using the manual support jack. The chamber was then purged with argon gas at a flow rate of 20 L/min for 30 minutes to fully eliminate residual humidity.

Each phosphoramidite and activator solution was stored in a pressurizable vial, and pressure control pumps were used to deliver these reagents to the inkjet printhead. During synthesis, the ink line pressure for each reagent was maintained at −0.5 kPa via the corresponding peristaltic pump to ensure consistent droplet ejection.

The synthesis cycle consisted of three major steps—detritylation, coupling, and oxidation—with intermediate washing steps between each. Prior to each coupling step, the substrate surface was dried by injecting argon gas through a solenoid valve. During the bulk solution steps (detritylation, washing, and oxidation), the substrate holder was positioned beneath the corresponding bulk solution nozzle. In the coupling step, the substrate passed under the inkjet printhead, which dispensed droplets of dT phosphoramidite and activator.

This cycle was repeated 14 times, resulting in the synthesis of a 14-mer poly(dT) sequence on the preloaded dT, yielding a final 15-mer poly(dT) oligonucleotide. Following synthesis, the substrate was immersed in concentrated ammonium hydroxide at 60°C for 2 hours to cleave the oligonucleotide from the solid support. The cleaved product was then precipitated using n-butanol and resuspended in deionized water for subsequent analysis.

## Urea-PAGE of synthesized oligonucleotides

40% acrylamide-bis solution (19:1), 10×TBE, urea, TEMED, and 10% ammonium persulfate solution (APS) were purchased from Biosesang (KR) and were used to prepare a 24% urea-PAGE gel for DNA analysis. TEMED and APS were added simultaneously to a mixture of the acrylamide solution, 10×TBE, and urea. After the solution was inverted 2–3 times, it was poured into the gel cassette to solidify. A warm 1×TBE solution was used as the running buffer to ensure clear band separation. Before loading the samples, each well was flushed to remove residual urea. Gel loading buffer II (Invitrogen) and the samples were mixed in a 1:1 ratio to a total volume of 10–20 µL. After electrophoresis, the gel was incubated in a 1×TBE solution containing SYBR Gold and then washed.

## HPLC analysis of oligonucleotide purity and coupling efficiency

Analysis of the synthesized oligonucleotides was performed on a ChroZen HPLC System (Young In Chromass, KR) equipped with a photodiode array (PDA) detector. Separation was achieved using an Agilent ZORBAX StableBond-C18 column (P/N 659950−702). The mobile phases consisted of (A) 5% acetonitrile (ACN) in 0.1 M triethylammonium acetate (TEAA, pH 7.0) and (B) 50% ACN in 0.1 M TEAA (pH 7.0).

The elution gradient was as follows: an initial isocratic hold at 5% B for 3 minutes, followed by a linear increase from 5% to 18% B over the next 26 minutes. The flow rate was maintained at 1.0 mL/min, and the column temperature was set to 60°C. The chromatogram was extracted at 260 nm for quantification. To analyze the coupling efficiency, the molar ratio for each oligonucleotide length was determined by dividing the corresponding peak area by the length of the oligonucleotide. As no capping step was employed during synthesis, the average coupling efficiency was then estimated by fitting this experimental molar distribution to a theoretical binomial distribution. The efficiency value that yielded the distribution most similar to the experimental data was determined as the average coupling efficiency.

## Results

### Overview of the OpenIDS2 system

We developed OpenIDS2, a second-generation inkjet-based oligonucleotide synthesizer, as an improved version of the previous OpenIDS platform. OpenIDS2 addresses limitations in device compactness, system integration, ease of fabrication, operational convenience, and reproducibility.

The system comprises six major components. (1) An inkjet printing module, (2) A bulk solution delivery system, (3) A compact synthesis chamber constructed from PMMA panels, (4) An integrated control panel based on open-source hardware and software, (5) Manual support jacks, and (6) A Raspberry Pi camera. An overview of the full system and the labeled configuration of these components is shown in Fig 1 and Fig 2.

The OpenIDS2 system consists of six functional components: (1) the inkjet printing system for dispensing amidite and activator solutions; (2) the bulk solution delivery system for DCA, ACN, and oxidizer; (3) a compact synthesis chamber housing a linear stage and substrate holder; (4) the integrated control panel for operating the printheads, stage, sensors, and pumps; (5) manual support jacks for vertical chamber actuation and sealing; and (6) a Raspberry Pi camera for print alignment.

Each component is labeled with a colored outline. Most structural parts were fabricated using laser-cut PMMA panels and 3D-printed parts.

OpenIDS2 was not designed as a large-scale commercial synthesizer for ultra-high-throughput production, but rather as a flexible, lab-scale platform optimized for the rapid and cost-effective fabrication of custom oligonucleotide arrays. The mechanical design was deliberately simplified, and aside from the electronic components, most of the structural elements were fabricated using laser-cut PMMA panels and 3D-printed parts. This approach lowers the technical barrier for fabrication and modification, enabling researchers with limited engineering experience to reproduce, adapt, and extend the system.

Large structural parts were printed using an FDM 3D printer with PLA, chosen for its low print difficulty and accessibility. Components requiring chemical resistance, such as the substrate holder, bulk solution nozzle holder, and pump housing, were printed using an LCD 3D printer with photopolymer resins.

The overall dimensions of the system frame are 580 × 440 × 300 mm, and its total volume has been reduced to approximately 35% of the previous version. The synthesis chamber measures 384 × 108 × 90 mm (approximately 3.7 L), which helps minimize inert gas consumption and purging time. The chamber, designed to accommodate only the linear stage and substrate holder, maximizes space efficiency by eliminating unnecessary internal structures. All control and mechanical modules are positioned outside the chamber to further reduce internal volume and stage travel distance.

The synthesis chamber is vertically raised and lowered using manual support jacks. The top lid includes an O-ring and is fixed to the frame, enabling a tight seal when the chamber is lifted. This mechanism allows users to load and retrieve substrates without tools, significantly enhancing operational convenience.

All 3D design files, including the printed structural components, are available for download via the GitHub repository. A detailed breakdown of component specifications and cost estimates is provided in S1 Table.

 

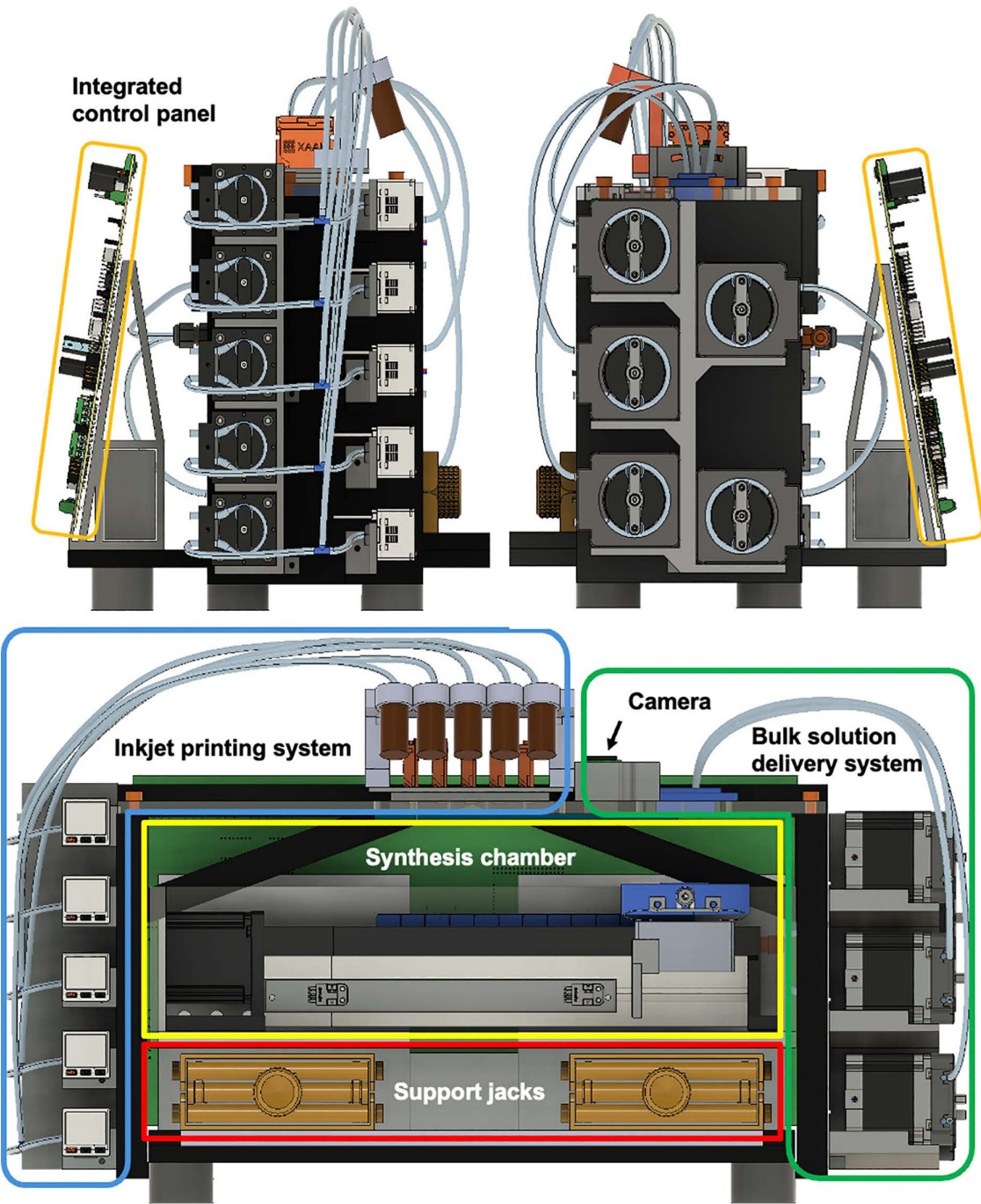

**Fig 1. Overview of the OpenIDS2 synthesizer and its major components.**

## Inkjet printing system

Five Xaar Irix printheads were mounted in a linear array at 18 mm intervals using a custom 3D-printed holder. Each print-head was connected to the ink supply system via a silicone tube. This ink supply system plays a critical role in maintaining

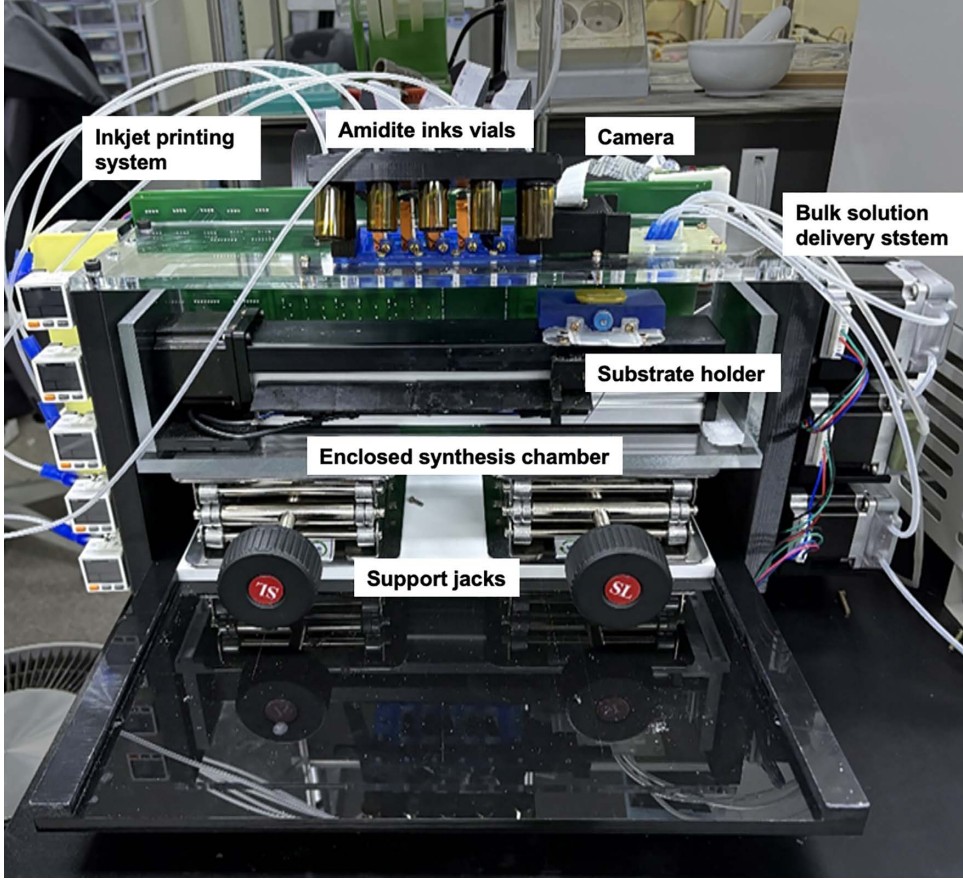

**Fig 2. Photograph of the fully assembled OpenIDS2 synthesizer.** The system includes the inkjet printing module, amidite ink vials, camera, bulk reagent delivery system, enclosed synthesis chamber, substrate holder (inside chamber), and support jacks.

consistent negative pressure, which is essential for stable droplet ejection in piezoelectric inkjet printing. To ensure reliable performance, precise regulation of backpressure is necessary to prevent backflow, bubble formation, and inconsistent ink refill. In our setup, the optimal backpressure was approximately –0.5 kPa, though this value varied depending on factors such as ink viscosity and the relative height between the vial and the printhead.

To achieve this precise pressure regulation, we developed a flexible ink supply system (Fig 3a). Each ink vial was connected to two tubes: one leading to the printhead inlet, and the other to a pressure control module comprising a peristaltic pump and a pressure sensor. When the pump rotated forward, it injected argon gas into the vial, thereby pushing ink toward the printhead. Reversing the pump direction allowed for pressure release or backflow adjustment. Internal vial pressure was continuously monitored in real time using the pressure sensor and was used as feedback to control the pump.

Each peristaltic pump assembly consisted of a NEMA17 stepper motor, a 3D-printed housing, bearings, and silicone tubing. Prior to synthesis, approximately 3 mL of ink was flushed through the printhead by pressurizing the ink vial, ensuring proper nozzle priming and removal of residual air. The pump then reversed to establish and maintain the desired negative pressure. As ink was gradually consumed during synthesis, the pump continued to inject argon gas to stabilize the internal pressure of the vial.

**a** **b**

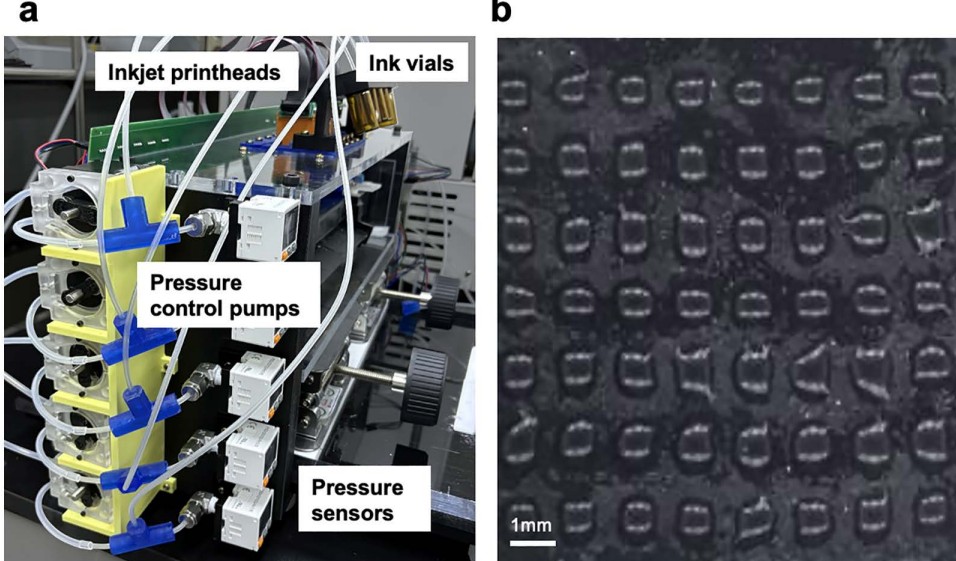

**Fig 3. Ink supply system and droplet characterization. (a)** Photograph of the ink supply and pressure control system. Each ink vial is connected to a dual-tube configuration: one tube delivers ink to the printhead, while the other is linked to a peristaltic pump and pressure sensor. The pump injects argon gas into the vial to control internal pressure and regulate ink flow. **(b)** Each droplet is successfully positioned with a 1.4 mm spacing, confirming compatibility of propylene carbonate-based reagents with the Xaar Irix printhead. Scale bar = 1 mm.

The inkjet printheads were used to dispense phosphoramidite monomers and activator solutions dissolved in PC. While ACN is traditionally used as a solvent in oligonucleotide synthesis, its high volatility often causes droplet instability and nozzle clogging, making it unsuitable for inkjet applications. In contrast, PC exhibits significantly lower volatility and has been widely adopted in inkjet-based DNA synthesis [20].

Using this setup, we successfully printed both phosphoramidite and activator solutions onto a CPG-arrayed substrate (Fig 3b). The droplets were clearly defined and uniformly deposited with a spacing of 1.4 mm, matching the patterned distribution of the CPG particles on the substrate. This precise and discrete droplet formation confirms successful inkjet deposition and demonstrates the compatibility of the PC-based reagent formulations with the printhead.

## Bulk solution delivery system

In our previous research, commercial syringe pumps were used to deliver bulk solutions. However, using three syringe pumps for detritylation, wash, and oxidation steps cost approximately $4,000, accounting for nearly 20% of the total device cost. To reduce cost and simplify fabrication for lab-scale applications, we replaced the syringe pumps with custom-built peristaltic pumps (Fig 4a). Unlike syringe pumps, which require space equivalent to the full stroke length of the syringe, the peristaltic pumps are compact and occupy only the volume of the motor itself. These pumps are composed of a NEMA 23 stepper motor, a 3D-printed housing, and a roller assembly, making them inexpensive and easy to reproduce via 3D printing.

The peristaltic pump offers several practical advantages. It facilitates maintenance, as tubing can be easily replaced. The pumped fluid never contacts any mechanical components, ensuring excellent chemical compatibility. Moreover, flow rate can be reliably controlled using only low-cost stepper motors, without requiring additional fluidic control hardware.

These improvements resulted in a significant reduction in overall system size, fabrication complexity, and cost. The total cost of building three peristaltic pumps for bulk solution delivery was less than $164.

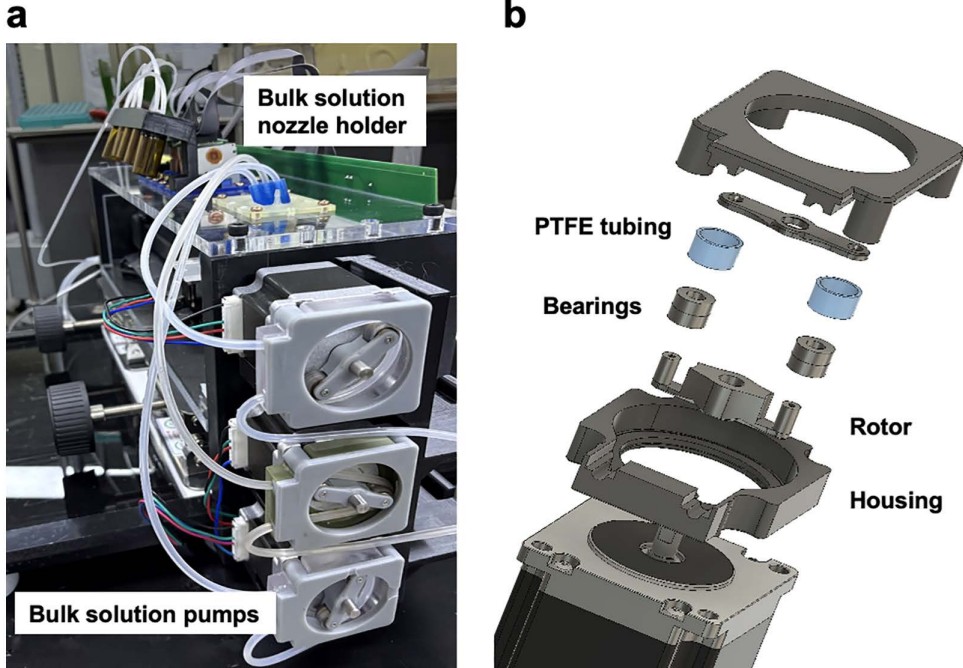

**Fig 4. Custom-built peristaltic pump for bulk solution delivery. (a)** Photograph of the assembled peristaltic pump used in OpenIDS2, consisting of a NEMA 23 stepper motor, a 3D-printed chemically resistant housing, and PP tubing. **(b)** Exploded CAD view showing the internal structure of the pump, including the modular housing, roller assembly, and bearing placement. The design enables easy assembly and maintenance while maintaining chemical resistance through the use of PTFE-wrapped bearings and LCD-printed resin parts.

Because bulk solutions for oligonucleotide synthesis contain aggressive chemicals, the chemical resistance of tubing and pump components was a critical design consideration. Conventional materials such as silicone or Tygon tubing were found to be unsuitable for long-term use with these reagents. Therefore, we selected polypropylene (PP) tubing for its superior chemical resistance. However, due to its stiffness, PP tubing required pre-softening prior to use by rotating the pump slowly or manually turning the rotor. To provide sufficient torque for stable operation, we used NEMA 23 stepper motors in combination with TMC2209 drivers operating at maximum rated current.

The roller assembly, which includes bearings and portions of the pump housing, comes into direct contact with the tubing. To ensure chemical durability in these parts, we printed the pump housing using photopolymer resin via LCD 3D printing, as many resins offer superior resistance to oligonucleotide synthesis reagents compared to FDM filaments. Furthermore, because commonly used carbon chrome bearing steel lacks adequate chemical resistance, we wrapped the miniature bearings in PTFE tubing to prevent corrosion and extend operational life. For ease of assembly and part replacement, the pump was designed as a modular 3D-printed structure comprising a main housing, bearing mounts, and a detachable roller arm (Fig 4b).

## Localized sealing mechanism for compact reaction space

OpenIDS2 was designed with a minimal overall volume, featuring an internal reaction chamber reduced to a compact volume of 384 × 108 × 90 mm (approximately 3.7 L). This compact configuration reduces the consumption of argon gas and contributes to overall miniaturization. However, in such a small reaction space, even minimal evaporation of volatile reagents can significantly elevate the local concentration of reactive substances, potentially compromising synthesis

quality. This issue is particularly critical in phosphoramidite chemistry, which is highly sensitive to moisture and prone to side reactions [21,22].

For example, the oxidation solution typically contains water and is volatile. If vapors from evaporated solutions diffuse into the chamber, they may contact phosphoramidite residing on the printhead nozzles. Such contact may hydrolyze phosphoramidites into inactive H-phosphonates, degrading synthesis performance [23]. Since each printhead dispenses only nanoliter volumes per cycle, even minor degradation at the nozzles can have significant effects on synthesis accuracy.

To mitigate this issue, OpenIDS2 incorporates a localized sealing mechanism. The substrate is mounted on a mobile holder attached to the linear stage. The holder includes a pair of bearings, and a 3D-printed arched flexible spring made from PP. These bearings slide along rigid guide rails attached to the top lid of the chamber. While the holder moves along this rail path, the spring remains compressed, exerting a constant upward force against the top lid (Fig 5a).

As the holder enters the bulk solution dispensing zone, the guiding rails terminate, allowing the spring to expand. This causes the holder to rise slightly and press its top surface against the bulk solution nozzle holder, forming a temporarily sealed reaction environment (Fig 5b). This sealing and unsealing mechanism is driven solely by the horizontal motion of the linear stage, and requires no additional actuators or sensors. Its mechanical simplicity enhances both system reliability and ease of fabrication.

Once the reaction microchamber is sealed, the bulk reagent is dispensed onto the substrate and retained for the designated incubation period to complete the chemical reaction (Fig 5c). After the reaction is complete, the residual bulk solution is aspirated through a suction inlet located below the reaction zone (Fig 5d). However, small amounts of reagent may remain on the substrate surface. To address this, once the chamber is unsealed, a directed stream of argon gas is blown across the substrate from above to remove any remaining bulk solution (Fig 5e).

Furthermore, to minimize vapor backflow through the nozzles, each peristaltic pump was programmed to briefly reverse after dispensing. This pulls any residual solution back into the tubing, preventing volatile reagents from lingering at the nozzle tip or diffusing into the chamber.

## Integration of control system via custom PCB architecture

One of the key technical improvements in OpenIDS2 is the integration of the entire electronic control system using a custom-designed PCB. This architecture significantly reduces wiring complexity while improving build consistency and ease of maintenance. Using the provided Gerber and BOM files, users can order the PCB from a fabrication service and assemble the system simply by connecting components through terminal blocks. This eliminates the need for manual wiring or breadboarding based on circuit diagrams, thereby lowering the fabrication barrier and improving reproducibility and operational reliability. A schematic overview of the control architecture, including all functional blocks and their interconnections, is provided (Fig 6a).

The Arduino Nanos, under master control, operate the linear stage motor, bulk solution pumps for reagent delivery and waste suction, solenoid valves, and limit sensors. The ink supply controller interfaces with five pressure sensors and five peristaltic pumps—one for each printhead—to regulate ink supply. This board continuously monitors and regulates the pressure of each channel to ensure stable ink delivery. The master Nano, ink supply controller, and printhead controllers communicate via I²C. The master communicates with the host PC via USB. The PCB was designed to accommodate Arduino Nanos via pin headers. In addition, the power supply lines for motor drivers and printheads include onboard voltage regulation and decoupling capacitors to minimize electrical noise and ensure stable operation (Fig 6b).

The PCB layout illustrates the physical arrangement of components on a two-layer FR4 substrate (Fig 7a). Due to the complexity of the schematic and layout, the complete project files, including editable source documents, are available in the project's GitHub repository (see Data Availability section). The fabricated board was mounted at the rear of the synthesizer and assembled by connecting all components via terminal blocks (Fig 7b).

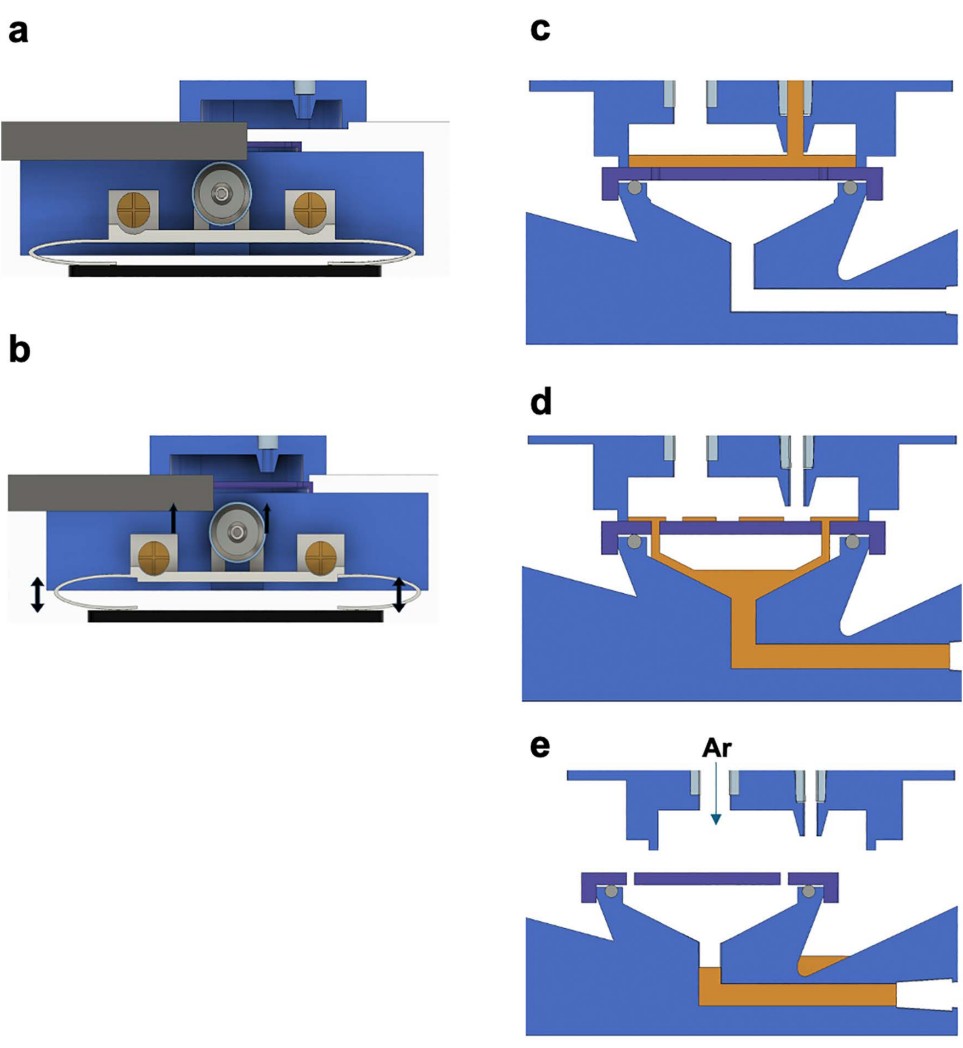

**Fig 5. Localized sealing mechanism for a compact reaction environment. (a)** While the substrate holder moves along the guide rail attached to the top lid, the PP spring inside the holder remains compressed, exerting upward force. **(b)** When the holder's bearings reach the bulk solution dispensing zone and the guide rail ends, the spring lifts the holder slightly, pressing it against the bulk solution nozzle holder to form a temporarily sealed reaction microchamber. **(c)** After the reaction microchamber is sealed, the bulk solution is dispensed onto the substrate and allowed to react for the designated duration. **(d)** Upon completion of the reaction, the residual bulk solution is aspirated through a suction inlet positioned below the reaction chamber. However, residual reagent may remain on the surface of the substrate. **(e)** After the chamber is unsealed, a directed stream of argon gas is blown across the substrate surface to remove residual bulk solution.

## Evaluation of OpenIDS2

To evaluate the operational capability of OpenIDS2, a 15-mer poly(dT) oligonucleotide was synthesized using standard phosphoramidite chemistry. First, urea-PAGE analysis was performed to confirm whether the oligonucleotide was synthesized to the target length. A dominant band corresponding to the target 15-mer poly(dT) product appeared, along with several shorter bands ranging from 11- to 14-mers (Fig 8a). This result suggests that the target sequence was successfully synthesized, indicating that the coupling efficiency was effective but not optimal.

For a more accurate analysis of the coupling efficiency, HPLC analysis was conducted (Fig 8b). The area percentage of the full-length product obtained from HPLC was 53.7%. Since the peak area is proportional to the product of the

a

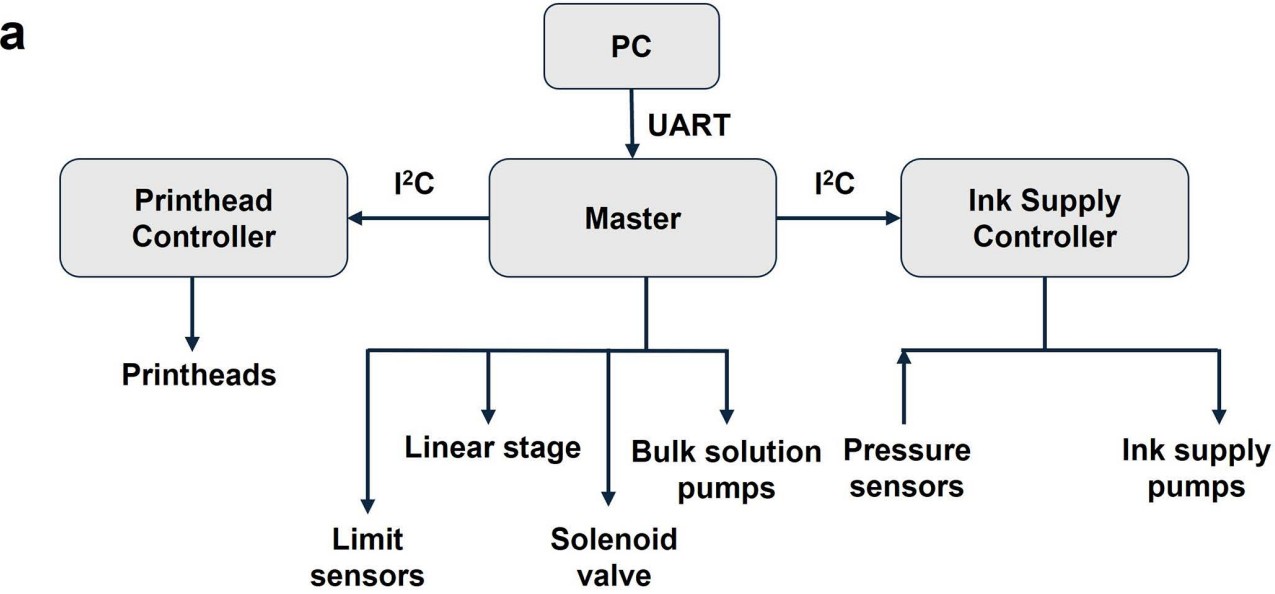

b

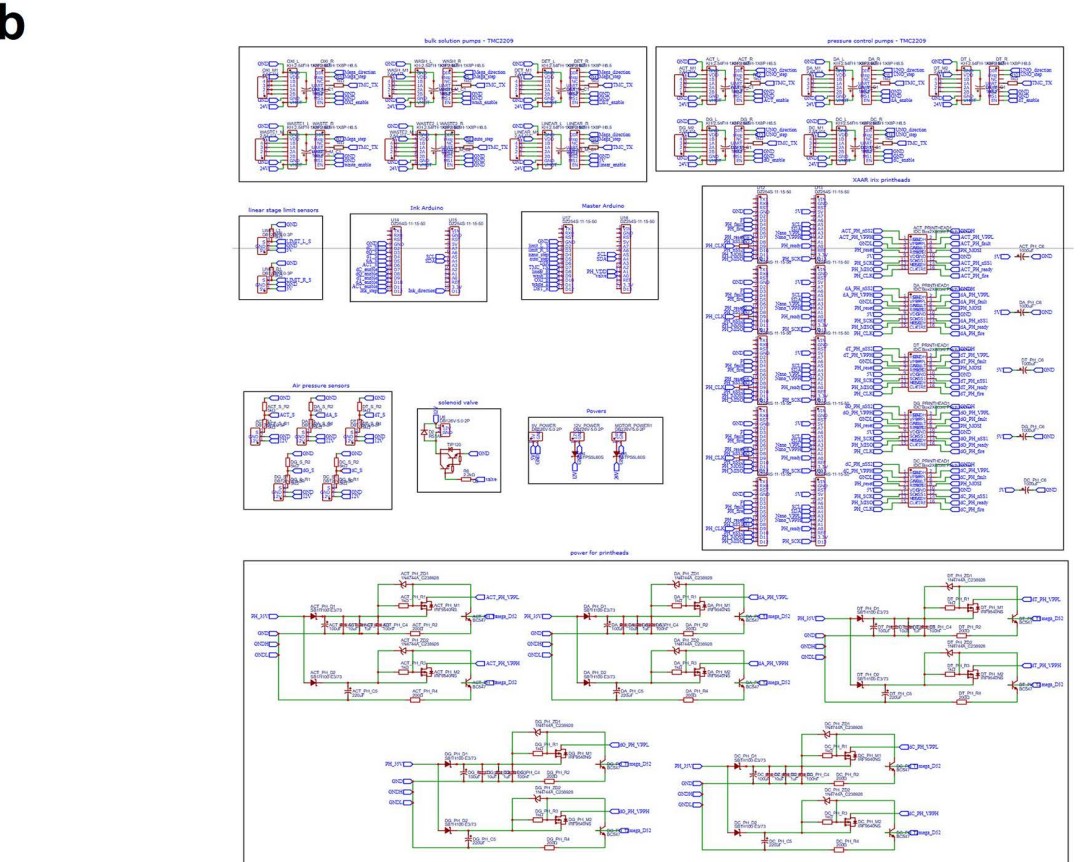

**Fig 6. PCB control system architecture and schematic. (a)** Block diagram of the control system architecture for OpenIDS2. The system consists of Arduino Nanos connected via I²C. The master controller manages the bulk solution pumps, solenoid valves, linear stage, and limit sensors. Ink supply controller manages pressure sensors and ink supply pumps, and printhead controllers manage printheads. The master controller communicates with the

host PC via USB. **(b)** Complete circuit schematic designed in EasyEDA, showing modules for motor control, pump drivers, pressure sensor interfaces, printhead circuits, and communication.

oligonucleotide's length and its molar amount, the area of each peak was divided by the corresponding nucleotide length to obtain the molar ratio. The resulting molar ratios for the 13-mer, 14-mer, and 15-mer products were approximately 7.1%, 32%, and 56.2%, respectively.

Based on this value, the estimated average per-step coupling efficiency was calculated to be approximately 96.1%. Despite the presence of minor byproducts, these results demonstrate the functional viability of the OpenIDS2 platform for multi-step oligonucleotide synthesis under typical laboratory conditions.

## Discussion

In this study, we presented OpenIDS2, a second-generation inkjet-based DNA synthesizer designed for laboratory-scale fabrication of custom oligonucleotide microarrays. Building on the foundation of the original OpenIDS platform, OpenIDS2 emphasizes enhanced reproducibility, ease of assembly, compactness, and accessibility as an open-source system.

The overall device volume was reduced to approximately one-third of its predecessor, and the synthesis chamber was redesigned as a localized, sealed environment to minimize inert gas usage and improve reaction stability. Structural components were fabricated using laser-cut PMMA panels and 3D-printed parts, allowing users to construct the system without specialized machining equipment.

By minimizing reliance on commercial components and implementing custom-designed modules, the total cost of building OpenIDS2 was reduced to approximately $4,000, which is only about 20% of the cost of OpenIDS1. Low-cost peristaltic pumps replaced syringe-based systems, reducing both cost and system complexity while improving maintainability. One of the most impactful improvements is the integration of the entire control system into a custom-designed PCB, which simplifies wiring, reduces assembly errors, and increases reproducibility. With all design files—including Gerber files, a bill of materials, and 3D CAD models—freely available, researchers can fabricate and assemble the device using only basic tools and widely available components. The system leverages open-source microcontrollers such as Arduino and Raspberry Pi, supporting straightforward customization and future upgrades.

Crucially, OpenIDS2 addresses common limitations encountered in open-source hardware. While many projects share designs, researchers often struggle to reproduce them due to unavailable components, outdated documentation, or reliance on specialized fabrication. By designing nearly all mechanical parts to be 3D-printable and selecting globally accessible electronics and materials, we aimed to minimize these barriers and enable reliable construction regardless of geographic or institutional constraints.

Beyond its function as a microarray synthesizer, this work contributes to the broader objective of establishing an initial open-source ecosystem for in-house instrument fabrication within the life sciences. In many life science fields, including synthetic biology and genomics, there is a steadily increasing demand for automation to enhance throughput and reduce experiment turnaround times. However, commercial instruments often present significant barriers to creating truly end-to-end automated workflows, as their closed architecture makes modification and seamless integration with other systems difficult. An ecosystem for custom, open-source hardware development can be a promising solution to this challenge.

However, many wet-lab biologists may still be reluctant to adopt such in-house systems. The effort required for hardware assembly, validation, and maintenance, coupled with the need for skills in engineering disciplines often unfamiliar to biologists, presents a considerable barrier. Nevertheless, much like the rapid growth of bioinformatics spurred many biologists to learn programming, we envision a future where the increasing importance of automation leads researchers to build their own hardware to scale and customize their experimental workflows. In this context, OpenIDS2 can serve as an excellent gateway for researchers entering the field of custom instrumentation. Its low cost and accessible fabrication

**a**

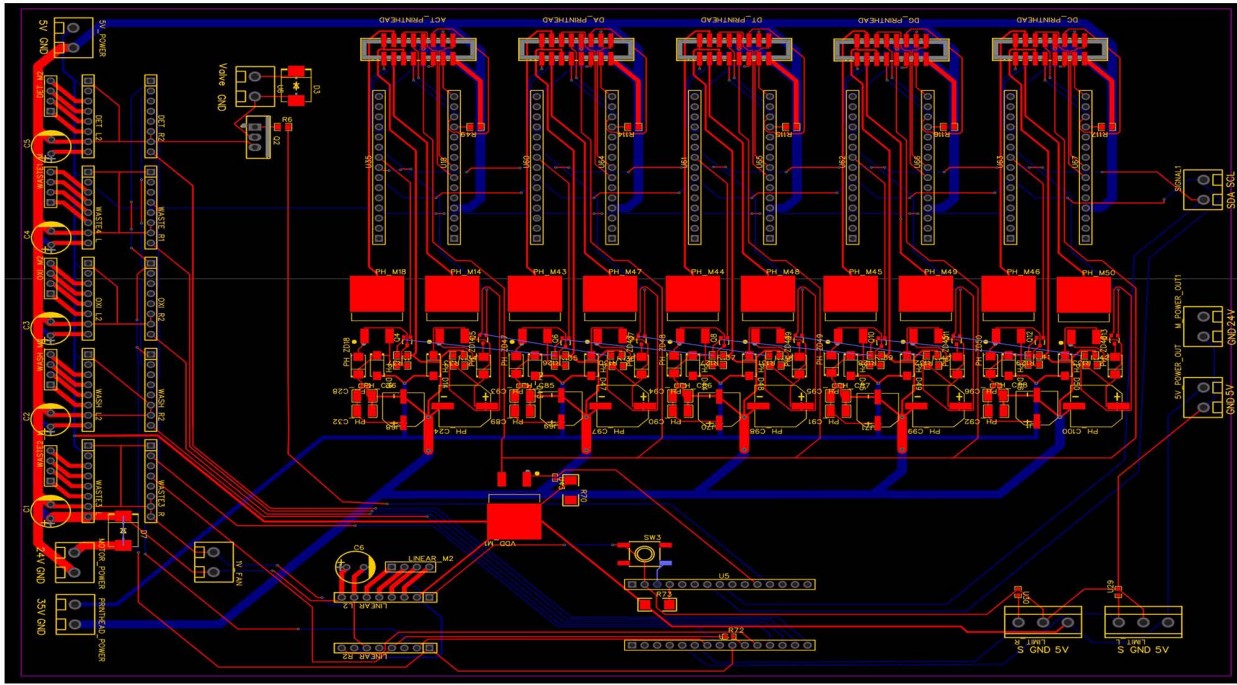

**b**

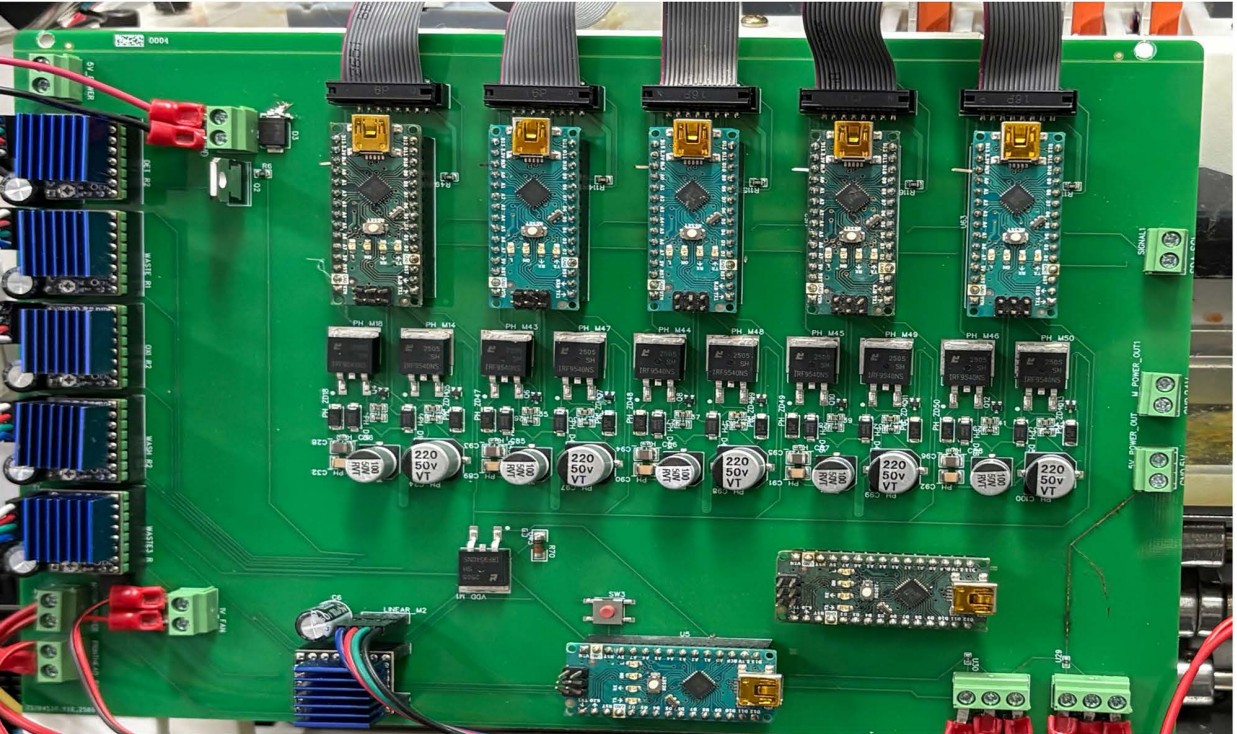

**Fig 7. PCB layout and fabricated control board. (a)** Two-layer PCB layout illustrating the routing and placement of key components, including stepper motor drivers, solenoid control, and inkjet printhead interface. **(b)** Photograph of the assembled control PCB, with Arduino Nanos mounted via pin

headers and external device ports connected. Terminal blocks allow quick and reproducible wiring. The board integrates all control functionalities in a compact and reproducible form.

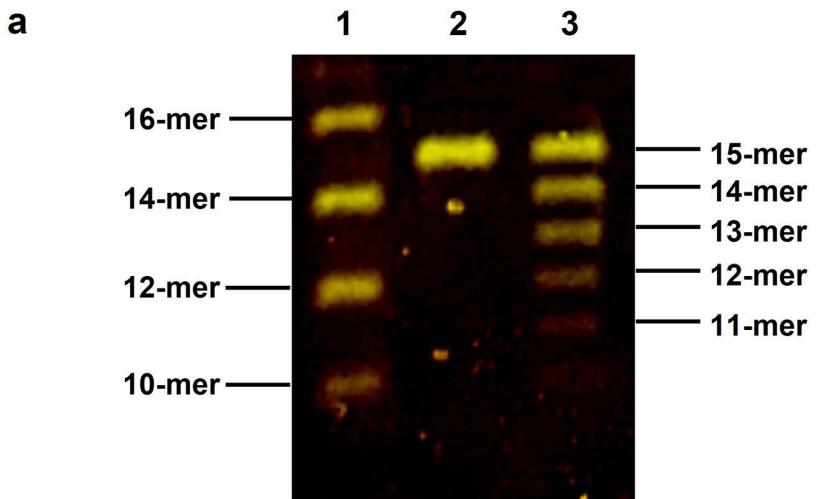

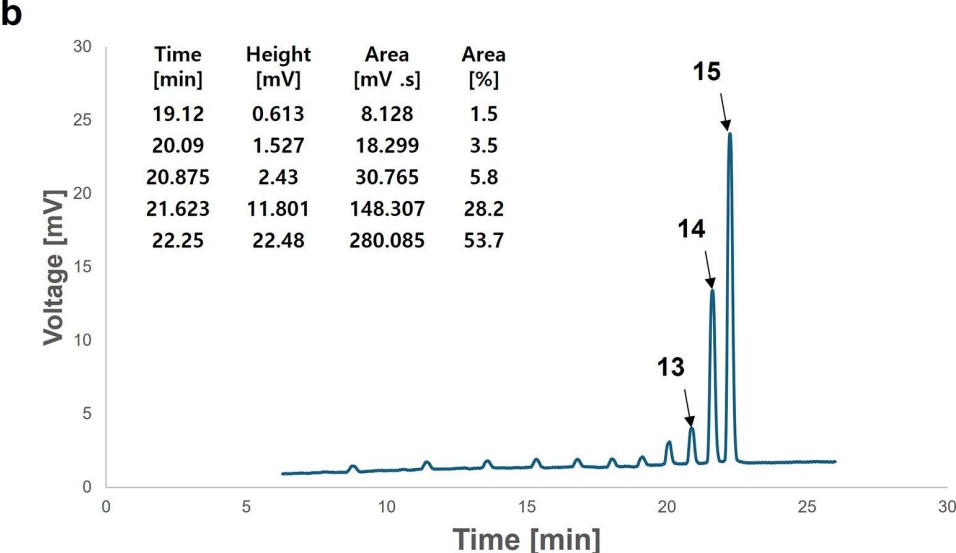

| Time [min] | Height [mV] | Area [mV .s] | Area [%] |
|---|---|---|---|
| 19.12 | 0.613 | 8.128 | 1.5 |
| 20.09 | 1.527 | 18.299 | 3.5 |
| 20.875 | 2.43 | 30.765 | 5.8 |
| 21.623 | 11.801 | 148.307 | 28.2 |
| 22.25 | 22.48 | 280.085 | 53.7 |

**Fig 8. Oligonucleotides synthesis for functional evaluation of the OpenIDS2. (a)** Urea-PAGE analysis of the synthesized poly(dT) 15-mer. Lane 1: DNA ladder; Lane 2: control 15-mer; Lane 3: synthesized sample. **(b)** HPLC analysis of the synthesized poly(dT) 15-mer. The stepwise coupling efficiency was estimated to be approximately 96.1% based on the peak areas. Consistent with the PAGE, peaks corresponding to several shorter oligonucletides were also observed. The chromatogram is displayed from 6 minutes onward to exclude the solvent front.

process lower the barrier to entry, making it not only a practical research tool but also a valuable educational platform for training the next generation of biologists in laboratory automation.

Ultimately, the core value of OpenIDS2 may lie not merely in being a low-cost, custom microarray synthesizer, but in its potential as a foundational platform that contributes to future innovation within the bio open-source community and can be modified and extended into diverse application-specific instruments. We believe that OpenIDS2 will serve as a

reproducible and extensible platform that lowers the barrier to entry for laboratory automation, thereby enabling new discoveries across various scientific fields.

All source files, including 3D models and circuit schematics, have been openly shared on GitHub.

## Supporting information

**S1 Table. Bill of materials for OpenIDS2.**
(XLSX)

## Author contributions

**Conceptualization:** Junhyeong Kim, Duhee Bang.

**Data curation:** Haeun Kim.

**Formal analysis:** Haeun Kim.

**Funding acquisition:** Duhee Bang.

**Investigation:** Junhyeong Kim.

**Methodology:** Junhyeong Kim.

**Project administration:** Duhee Bang.

**Software:** Junhyeong Kim.

**Supervision:** Duhee Bang.

**Visualization:** Junhyeong Kim.

**Writing – original draft:** Junhyeong Kim, Haeun Kim.

**Writing – review & editing:** Junhyeong Kim, Haeun Kim, Duhee Bang.

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
