## [Decision Letter · Decision Letter 0]

4 Sep 2025

Dear Dr. Bang,

Thank you for submitting your manuscript to PLOS ONE. After careful consideration, we feel that it has merit but does not fully meet PLOS ONE’s publication criteria as it currently stands. Therefore, we invite you to submit a revised version of the manuscript that addresses the points raised during the review process.

We look forward to receiving your revised manuscript.

Kind regards,

Dola Sundeep

Academic Editor

PLOS ONE

Journal Requirements:

This work was supported by a National Research Foundation of Korea (NRF) grant funded by the Korean Ministry of Science and ICT (NRF-2022M3C1A3081366), and by the National Research Foundation of Korea (NRF) grant funded by the Korea government (MSIT) (RS-2024-00338316).

This work was supported by a National Research Foundation of Korea (NRF) grant funded by the Korean Ministry of Science and ICT (NRF-2022M3C1A3081366), and by the National Research Foundation of Korea (NRF) grant funded by the Korea government (MSIT) (RS-2024-00338316)

This work was supported by a National Research Foundation of Korea (NRF) grant funded by the Korean Ministry of Science and ICT (NRF-2022M3C1A3081366), and by the National Research Foundation of Korea (NRF) grant funded by the Korea government (MSIT) (RS-2024-00338316).

Additional Editor Comments :

Dear Professor Duhee Bang,

We have now received the required number of reviewer comments for your manuscript (Manuscript Number: PONE-D-25-42242) entitled "OpenIDS2: A low-cost, 3D-printed, open-source platform for reproducible construction of DNA microarray synthesizers" submitted to PLOS ONE.

Based on the reviewers’ evaluations, the decision of the Editorial Board is Major Revisions. We kindly request you to revise your manuscript in accordance with the reviewers’ comments and resubmit for further consideration.

Thank you for choosing PLOS ONE as a venue for your work.

Sincerely,

Dr. Dola Sundeep

Academic Editor

PLOS ONE

Reviewers' comments:

Reviewer's Responses to Questions

**Comments to the Author**

1. Is the manuscript technically sound, and do the data support the conclusions?

Reviewer #1: Partly

Reviewer #2: Yes

Reviewer #3: Yes

2. Has the statistical analysis been performed appropriately and rigorously?

Reviewer #1: N/A

Reviewer #2: Yes

Reviewer #3: N/A

3. Have the authors made all data underlying the findings in their manuscript fully available?

Reviewer #1: Yes

Reviewer #2: Yes

Reviewer #3: Yes

4. Is the manuscript presented in an intelligible fashion and written in standard English?

Reviewer #1: Yes

Reviewer #2: Yes

Reviewer #3: Yes

Reviewer #1: The manuscript describes the development of OpenIDS2, an open-source, inkjet-based DNA synthesizer utilizing largely 3D-printed components and an integrated PCB for control. While the engineering improvements and commitment to open-source principles are appreciated, I have concerns regarding the practical relevance and adoption potential of such a system in today’s context.

Major Comments:

1.Technological Relevance: The field of oligonucleotide synthesis has matured significantly, with numerous commercial providers offering high-quality, low-cost, and rapid services. Custom in-house synthesis—especially via microarrays—has become increasingly niche. The authors should better clarify what unique advantages OpenIDS2 offers over existing commercial solutions, particularly in terms of cost, flexibility, or accessibility, and provide convincing evidence or use-cases that would motivate labs to adopt this platform.

2.Application Gap: Microarray-based applications (e.g., hybridization arrays, SNP detection) have largely been supplanted by sequencing-based technologies. The authors should expand on how an open-source synthesizer like OpenIDS2 could enable novel applications not easily achievable with commercial instruments—e.g., in synthetic biology, DNA data storage, bespoke diagnostic arrays, or education—to enhance the significance of their work.

3.Practical Adoption: Although the system is designed to be reproducible and low-cost, the effort required to assemble, validate, and maintain the hardware may deter many wet-lab biologists. The discussion would benefit from a realistic assessment of the target user base and the types of settings (e.g., teaching labs, remote laboratories) where OpenIDS2 could offer tangible advantages.

4.Future Value: Given the pace of advancement in synthetic biology and genomics, the authors might consider emphasizing the platform’s potential for customization and adaptation rather than positioning it primarily as a tool for conventional microarray synthesis. If possible, demonstrating a novel application—or integration with emerging methodologies—would greatly strengthen the manuscript.

5.Recommendation: The manuscript may require major revisions to better articulate its unique value proposition and applicability in contemporary research settings. Should the authors adequately address these points, this work could represent a valuable contribution to the open-source biology community. Alternatively, the innovation may be better suited for patent protection given its engineering focus, should substantial biological applicability remain unclear.

Reviewer #2: In this paper, a second-generation inkjet-based DNA synthesizer (OpenIDS2) was introduced. Comparing to the first generation (OpenIDS platform), it is much convenient to assemble, and the cost is much lower.

The major question for this paper is that the coupling efficiency is relatively low (approximately 87%). The coupling efficiency in DNA microarray fabrication typically ranges between 98.5% and 99.9%. The author should discuss further about this problem. What is the reason causing low coupling efficiency, and how to improve the coupling efficiency. The quality of oligo is the most important part in DNA oligonucleotide synthesis. As show in Fig 8, the full length product (15mer) was only ~10% of the total synthesized oligo.

In Figure 8, which method was used to visualize DNA oligo in the gel? How to calculate the coupling efficiency? Detail experimental information should be added.

Reviewer #3: The authors present a compact open-source oligonucleotide synthesizer. While the OpenIDS2 instrument was thoughtfully designed, the stepwise coupling efficiency of 87% and the 14% final yield of a 15-mer oligo are quite disappointing, are not useful in real-life oligo synthesis scenarios, and are significantly inferior to what was reported from their first-generation instrument. If the authors can acknowledge in their manuscript that the new OpenIDS2 is a lighter-weight but lower-performance alternative to their original OpenIDS, then I would support publishing in PLOS ONE.

**Do you want your identity to be public for this peer review?** For information about this choice, including consent withdrawal, please see our Privacy Policy

Reviewer #1: No

Reviewer #2: **Yes: ** Gaofei Lu

Reviewer #3: No

---

## [Author Response · Author response to Decision Letter 1]

27 Oct 2025

Author’s response for reviewer 1

Comment 1

Technological Relevance: The field of oligonucleotide synthesis has matured significantly, with numerous commercial providers offering high-quality, low-cost, and rapid services. Custom in-house synthesis—especially via microarrays—has become increasingly niche. The authors should better clarify what unique advantages OpenIDS2 offers over existing commercial solutions, particularly in terms of cost, flexibility, or accessibility, and provide convincing evidence or use-cases that would motivate labs to adopt this platform.

Author’s response

We thank the reviewer for their valuable feedback. We fully agree with the point that it is crucial to clearly articulate the unique value of the OpenIDS2 platform, especially in an era where commercial oligonucleotide synthesis services are prevalent.

In response to this feedback, we have revised the Introduction section to further emphasize and elaborate on the specific advantages that OpenIDS2 offers compared to commercial solutions. We have now detailed the distinct benefits of performing in-house oligonucleotide array synthesis with our platform. These include: 1) A significant reduction in overall project lead times. We clarify that while commercial services also provide rapid synthesis, the total turnaround time is often dominated by shipping and logistical delays. Our in-house platform eliminates this bottleneck, enabling truly rapid prototyping and iterative experimentation. 2) Complete process control, which enables research using non-standard reagents or unconventional substrates. 3) The elimination of the need to transmit proprietary or sensitive sequence data to a third-party vendor.

We believe these revisions will help readers to more clearly understand the specific research environments and objectives for which OpenIDS2 is suitable, thereby providing a stronger motivation for the adoption of our platform.

Comment 2

Application Gap: Microarray-based applications (e.g., hybridization arrays, SNP detection) have largely been supplanted by sequencing-based technologies. The authors should expand on how an open-source synthesizer like OpenIDS2 could enable novel applications not easily achievable with commercial instruments—e.g., in synthetic biology, DNA data storage, bespoke diagnostic arrays, or education—to enhance the significance of their work.

Author’s response

We thank the reviewer for their insightful suggestion. We fully agree that it is essential to emphasize the potential of OpenIDS2 for novel applications in modern life sciences—beyond the limitations of commercial instruments—to enhance the significance of our work. We have addressed this point in combination with the advice from Comment 4, which recommended that we "consider positioning this platform not just as a conventional microarray synthesis tool, but to emphasize its custom development and applicability." Accordingly, we have revised the Discussion section to specifically present the new possibilities that OpenIDS2 can unlock.

Comment 3

Practical Adoption: Although the system is designed to be reproducible and low-cost, the effort required to assemble, validate, and maintain the hardware may deter many wet-lab biologists. The discussion would benefit from a realistic assessment of the target user base and the types of settings (e.g., teaching labs, remote laboratories) where OpenIDS2 could offer tangible advantages.

Author’s response

We thank the reviewer for their comment from a practical standpoint. We acknowledge that despite our efforts to simplify the design, OpenIDS2 is not a 'plug-and-play' instrument and does require a certain level of effort and engineering knowledge for its assembly and maintenance. In the Discussion section, we acknowledge this challenge, stating that many wet-lab biologists may currently be reluctant to adopt such in-house systems.

However, we also discuss the increasing necessity for open-source laboratory instruments within the broader trajectory of advancements in the life sciences, positioning our research as a foundational work in this emerging trend. Furthermore, we have highlighted that our platform can serve as a valuable educational tool for training the next generation of biologists in the field of laboratory automation.

Comment 4

Future Value: Given the pace of advancement in synthetic biology and genomics, the authors might consider emphasizing the platform’s potential for customization and adaptation rather than positioning it primarily as a tool for conventional microarray synthesis. If possible, demonstrating a novel application—or integration with emerging methodologies—would greatly strengthen the manuscript.

Author’s response

We deeply appreciate the reviewer's forward-looking advice. We fully concur with the opinion that the true value of OpenIDS2 lies in its potential as a 'customizable and extensible platform' rather than in its perfection as a standalone device. Indeed, this aligns perfectly with the primary objective of our research.

Our work on the microarray synthesizer serves as the starting point for a larger ambition: to establish an end-to-end automated experimental environment within an academic laboratory setting. The choice of the inkjet method was deliberate, as it serves as a foundational technology capable of dispensing diverse materials beyond amidites—such as cells or labeling agents—to precise locations, thus enabling a broad range of micro-scale experiments.

To better highlight this perspective, we have revised the tone of the Discussion section to place greater emphasis on the platform's extensibility and adaptability. Specifically, we have underscored that achieving end-to-end automation ultimately necessitates custom instrumentation and have highlighted that, "In this context, OpenIDS2 can serve as an excellent gateway for researchers entering the field of custom instrumentation."

Through these revisions, we believe we now more effectively convey that OpenIDS2 is not intended as a single-purpose tool, but as a living, open-source platform poised to evolve and be adapted for diverse future research applications with community involvement.

Comment 5

Recommendation: The manuscript may require major revisions to better articulate its unique value proposition and applicability in contemporary research settings. Should the authors adequately address these points, this work could represent a valuable contribution to the open-source biology community. Alternatively, the innovation may be better suited for patent protection given its engineering focus, should substantial biological applicability remain unclear.

Author’s response

We are deeply grateful for the reviewer's comprehensive review and constructive suggestions. In accordance with the recommendations spanning Comments 1-4, we have revised much of the content in the Introduction and Discussion sections, and have oriented the primary value of this research as a contribution to the open-source biology community. 

Author’s response for reviewer 2

Comment 1

The coupling efficiency is relatively low (approximately 87%). The coupling efficiency in DNA microarray fabrication typically ranges between 98.5% and 99.9%. The author should discuss further about this problem. What is the reason causing low coupling efficiency, and how to improve the coupling efficiency. The quality of oligo is the most important part in DNA oligonucleotide synthesis.

Author’s response

We fully agree with the reviewer's comment. The low coupling efficiency was indeed the most significant issue for us. Since the initial manuscript submission, we have made numerous attempts to improve this efficiency. In conclusion, we have successfully increased the coupling efficiency to 96.1%.

The primary factor that lowered the efficiency was unstable jetting. We intermittently observed that some of the 128 nozzles dispensed insufficient ink volume or failed to jet at all. These spots, where little to no ink was dispensed, resulted in minimal synthesis during the corresponding cycle, which significantly lowered the average coupling efficiency. The major reason was data corruption during transmission due to noise, originating from a design issue in the PCB. This problem was resolved by modifying the circuit design.

The second reason was related to the miniaturization of the synthesizer, which resulted in a smaller synthesis chamber and placed the printhead closer to the bulk solution dispensing area. During the oxidation step, the highly volatile, water-containing oxidation reagent could more easily contaminate the small synthesis chamber with moisture. Consequently, the amidite and activator solutions waiting at the printhead nozzles were exposed to this moisture. To solve this problem, we changed the direction of the inert gas inlet and continuously drew the internal air from the substrate holder using the waste pump. This established a flow of inert gas from the printhead towards the bulk solution dispensing area, allowing us to increase the coupling efficiency to 96.1%.

Although this efficiency is still lower than that of commercial instruments and requires further investigation and improvement, we believe that we can reach the performance level of commercial equipment. Our primary goal in designing our synthesizer was to make it easily reproducible for anyone and to disclose all design details. If OpenIDS is actually used by other laboratories and undergoes continuous optimization by the community, as we intended, we are confident it will achieve commercial-grade performance.

Reflecting the points above, we have revised Figures 6-8 in the Results section and added information regarding the inert gas flow to the 'Synthesizer construction' subsection in the Materials and Methods section.

Comment 2

In Figure 8, which method was used to visualize DNA oligo in the gel? How to calculate the coupling efficiency? Detail experimental information should be added.

Author’s response

We thank the reviewer for this helpful comment. We have added a new subsection titled "Urea-PAGE of synthesized oligonucleotides" to the Materials and Methods section. Furthermore, to enable a more precise calculation of the coupling efficiency, we have incorporated analysis via HPLC. The corresponding HPLC data has been added as Figure 8b, and a new subsection detailing this process, titled "HPLC analysis and calculation of DNA coupling efficiency," has also been added to the Materials and Methods section

Author’s response for reviewer 3

Comment

The stepwise coupling efficiency of 87% and the 14% final yield of a 15-mer oligo are quite disappointing, are not useful in real-life oligo synthesis scenarios, and are significantly inferior to what was reported from their first-generation instrument.

Author’s response

We thank the reviewer for their in-depth review. We fully concur that the initial 87% efficiency was insufficient for practical applications, and its performance gap with the first-generation instrument was a critical issue we were determined to resolve. We are pleased to report that our subsequent optimization efforts have yielded significant improvements. The synthesizer now achieves a step-wise coupling efficiency of 96.1%, resulting in a final yield of approximately 56.2% for a 15-mer oligo

The primary factor that lowered the efficiency was unstable jetting. We intermittently observed that some of the 128 nozzles dispensed insufficient ink volume or failed to jet at all. These spots, where little to no ink was dispensed, resulted in minimal synthesis during the corresponding cycle, which significantly lowered the average coupling efficiency. The major reason was data corruption during transmission due to noise, originating from a design issue in the PCB. This problem was resolved by modifying the circuit design.

The second reason was related to the miniaturization of the synthesizer, which resulted in a smaller synthesis chamber and placed the printhead closer to the bulk solution dispensing area. During the oxidation step, the highly volatile, water-containing oxidation reagent could more easily contaminate the small synthesis chamber with moisture. Consequently, the amidite and activator solutions waiting at the printhead nozzles were exposed to this moisture. To solve this problem, we changed the direction of the inert gas inlet and continuously drew the internal air from the substrate holder using the waste pump. This established a flow of inert gas from the printhead towards the bulk solution dispensing area, allowing us to increase the coupling efficiency to 96.1%.

Reflecting the points above, we have revised Figures 6-8 in the Results section and added information regarding the inert gas flow to the 'Synthesizer construction' subsection in the Materials and Methods section.

---

## [Decision Letter · Decision Letter 1]

25 Nov 2025

OpenIDS2: A low-cost, 3D-printed, open-source platform for reproducible construction of DNA microarray synthesizers

PONE-D-25-42242R1

Dear Dr. Duhee Bang,

We’re pleased to inform you that your manuscript has been judged scientifically suitable for publication and will be formally accepted for publication once it meets all outstanding technical requirements.

Kind regards,

Dola Sundeep

Academic Editor

PLOS ONE

Additional Editor Comments (optional):

Dear Duhee Bang,

We are pleased to inform you that we have now received the required number of reviewer comments for your manuscript titled PONE-D-25-42242R1 – “OpenIDS2: A low-cost, 3D-printed, open-source platform for reproducible construction of DNA microarray synthesizers”, submitted to PLOS ONE.

Based on the reviewers’ comments and evaluations, the Editorial Board is pleased to ACCEPT your manuscript for publication.

Thank you for your valuable contribution.

Warm regards,

Dr. Dola Sundeep

Academic Editor

Reviewers' comments:

Reviewer's Responses to Questions

**Comments to the Author**

Reviewer #1: All comments have been addressed

Reviewer #2: All comments have been addressed

2. Is the manuscript technically sound, and do the data support the conclusions?

Reviewer #1: Yes

Reviewer #2: Yes

3. Has the statistical analysis been performed appropriately and rigorously?

Reviewer #1: Yes

Reviewer #2: Yes

4. Have the authors made all data underlying the findings in their manuscript fully available?

Reviewer #1: No

Reviewer #2: Yes

5. Is the manuscript presented in an intelligible fashion and written in standard English?

Reviewer #1: Yes

Reviewer #2: Yes

Reviewer #1: The authors should make all data underlying the findings described in the manuscript fully available without restriction, with rare exception (please refer to the Data Availability Statement in the manuscript PDF file).

Reviewer #2: (No Response)

**Do you want your identity to be public for this peer review?** For information about this choice, including consent withdrawal, please see our Privacy Policy

Reviewer #1: **Yes: ** Shunqi Wang

Reviewer #2: No

---

## [Editor Report · Acceptance letter]

PONE-D-25-42242R1

PLOS One

Dear Dr. Bang,

I'm pleased to inform you that your manuscript has been deemed suitable for publication in PLOS One. Congratulations! Your manuscript is now being handed over to our production team.

Kind regards,

on behalf of

Dr. Dola Sundeep

Academic Editor

PLOS One